# Intramuscular Fat Prediction Using Color and Image Analysis of Bísaro Pork Breed

**DOI:** 10.3390/foods10010143

**Published:** 2021-01-12

**Authors:** Alfredo Teixeira, Severiano R. Silva, Marianne Hasse, José M. H. Almeida, Luis Dias

**Affiliations:** 1Mountain Research Centre (CIMO), Escola Superior Agrária/Instituto Politécnico de Bragança, Campus Sta Apolónia Apt 1172, 5301-855 Bragança, Portugal; mariannehassehasse@gmail.com (M.H.); ldias@ipb.pt (L.D.); 2Veterinary and Animal Research Centre (CECAV), Universidade Trás-os-Montes e Alto Douro, Quinta de Prados, 5000-801 Vila Real, Portugal; ssilva@utad.pt (S.R.S.); malmeida@utad.pt (J.M.H.A.)

**Keywords:** intramuscular fat, prediction, image analysis, Bísaro pork

## Abstract

This work presents an analytical methodology to predict meat juiciness (discriminant semi-quantitative analysis using groups of intervals of intramuscular fat) and intramuscular fat (regression analysis) in Longissimus thoracis et lumborum (LTL) muscle of Bísaro pigs using as independent variables the animal carcass weight and parameters from color and image analysis. These are non-invasive and non-destructive techniques which allow development of rapid, easy and inexpensive methodologies to evaluate pork meat quality in a slaughterhouse. The proposed predictive supervised multivariate models were non-linear. Discriminant mixture analysis to evaluate meat juiciness by classified samples into three groups—0.6 to 1.1%; 1.25 to 1.5%; and, greater than 1.5%. The obtained model allowed 100% of correct classifications (92% in cross-validation with seven-folds with five repetitions). Polynomial support vector machine regression to determine the intramuscular fat presented R^2^ and RMSE values of 0.88 and 0.12, respectively in cross-validation with seven-folds with five repetitions. This quantitative model (model’s polynomial kernel optimized to degree of three with a scale factor of 0.1 and a cost value of one) presented R^2^ and RSE values of 0.999 and 0.04, respectively. The overall predictive results demonstrated the relevance of photographic image and color measurements of the muscle to evaluate the intramuscular fat, rarther than the usual time-consuming and expensive chemical analysis.

## 1. Introduction

Imaging analysis techniques have been used intensively in animal science to predict body composition, carcass grading and to assess meat quality traits. Particularly, the use of online non-invasive and non-destructive techniques, avoiding the carcass dissections or chemical analysis, have become more and more interesting for the pork meat industry [1,2]. In meat studies, attention has been focused on the relationships between the amount of intramuscular fat and the tenderness, juiciness, flavor and palatability, since these parameters have been highlighted as essential to consumer acceptability [3]. The visual aspect of intramuscular fat, commonly called marbling, associated with the color of the meat, are the main factors in the consumer’s purchase decision, as they are the only ones that can be accessed at the time of purchase, once tenderness, juiciness and flavor are only possible when cooking and tasting. A study to evaluate the credence cues of pork are more important than consumers’ culinary skills to boost their purchasing intention showed that marbled increased by 12% the expressed willingness to pay [4]. The marbling is more visible in beef than in pork except in some much-marbled genotypes [5] as well as in the meat of older animals, once the intramuscular fat develops later than the other fat depots [6]. Intramuscular fat determination is done by chemical analysis while marbling assessment is usually done visually using different reference standards approved by the various organizations and councils of pork producers or by the competent entity in each country. However, the method based on the visual appraisal has the disadvantage of being subjective, and the chemical analysis is expensive, laborious and time-consuming. To overcome those issues, the use of spectroscopic and image-based techniques to assess marbling in pigs was proposed as a feasible and accurate approach [2,7]. In order to perform a rapid online grading of pork marbling degree, several techniques have been developed—near-infrared reflectance spectroscopy [8,9]; nuclear magnetic resonance [10]; computed tomography [11,12]; hyperspectral imaging [13,14]; and computer image analysis [15,16]. Although some of those techniques have been proposed, particularly in Europe and specifically in the oldest marbled breeds such as Bísaro, there is a lack of a reliable, objective and inexpensive system to assess the intramuscular fat in pork. Moreover, the Bisaro pork (Celtic type), raised mainly in the north of Portugal, has shown an increasing interest in the production of premium meat products, having two types of carcasses with protected designation of meat products (DOP) [17]. Specifically, a piglet (Bísaro piglet) weighing up to 12 kg of carcass and pigs with more than 60 kg of carcass, which supplies the fresh meat market and/or the meat industry, for the production of ham or sausage. In order to obtain meat products from the Bisaro pig with greater value, it is essential to have quick analytical tools to control the quality of the meat, such as the intramuscular fat content, which is essential to guarantee the quality of differentiated meat products.

Our study aimed to predict meat juiciness (discriminant semi-quantitative analysis using groups of intervals of intramuscular fat) and intramuscular fat (regression analysis) in Longissimus thoracis et lumborum (LTL) muscle of Bísaro pigs using color and image analysis (IA), for the purpose of developing rapid, easy and inexpensive methodologies to evaluate pork meat quality in a slaughterhouse.

## 2. Material and Methods

To carry out the present work, a consortium was created between the National Association of Bísaro Pork Breeders, an industrial unit for processing pork (Bisaro Salsicharia Tradicional) and a research center (Laboratory of Meat Carcass and Meat Quality at Agrarian School of Polytechnic Institute of Bragança) and was part of the BISIPORC project, financed by the PRODER program, measure 4.1 Cooperation for Innovation. The animals were raised, selected and supplied by the Breeder Association, slaughtered at Municipal Slaughtered House of Bragança and the carcasses sent to the meat industry and to the meat quality laboratory for study and processing.

### 2.1. Animals and Sampling

Muscle images analyzed were from 20 Bísaro pork carcasses (10 males and 10 females) slaughtered at the slaughterhouse in Bragança, Portugal, in compliance with the European Rules [18] for the protection of animals. Body weight varied between 22 and 111 kg. Feeding system of the piglets, the slaughter procedure and carcass fabrication were previously described by Álvarez-Rodríguez and Teixeira (2019) [6]. After slaughter, the carcass weight (CW) was recorded, then the carcass was placed in a cooling chamber. After cooling at 4 °C for 24 h, the cold carcass weight was recorded (CCW). Then, the carcasses were carefully halved, and a sample of the cutlets between the 12–13th ribs was taken from the left side of the carcass.

### 2.2. Color Measurement

Over the LTL, the color was measured using a Minolta CM-2006d spectrophotometer (Konica Minolta Holdings, Inc., Osaka, Japan) in CIEL*a*b* space [19]. The L* for the lightness from black (0) to white (100), a* from green (−) to red (+), and b* from blue (−) to yellow (+). 

The angle (h*ab* = tan^−1^$\left(\frac{b^{*}}{a^{*}}\right)$ and chroma $C_{ab}^{*}=\sqrt{\left(a^{*}\right){\;}^{2}+{\left(b^{*}\right)}^{2}}\;$ were also calculated. These five parameters from CIELAB color were measured three times at different aleatory points of the muscle.

### 2.3. Chemical Analysis

The cutlets from the right side of the same animals were taken to chemical analysis. For the determination of intramuscular fat, the LTL muscle was separated from the bone and cleaned of subcutaneous and intermuscular fat. The LTL samples were minced using an Ultra Turrax homogenizer (Ultra Turrax T25, IKA, Staufen, Germany). The chemical IMF content of LTL samples was obtained after ether extraction in a Tecator Soxtec HT 1043 (Höganäs, Sweden), and was determined gravimetrically, after evaporating the petroleum ether solvent according to the AOAC [20] method.

### 2.4. Image Acquisition

In preparing the sample cutlets that were used for image acquisition, care was taken to cut them all 3 cm thick. For this, a band saw was used, equipped with guides that allowed the cutting accuracy. The cutlets were placed over a platform with an opaque black background. This background has the purpose of minimizing backlight. It was also placed a scale for tissues and IMF features measurements. To capture images, the system consisted of an Olympus EM-5 digital photo camera (Olympus, Tokyo, Japan) with a 16 Megapixel sensor that was mounted vertically on a support, at a distance of 30 cm from the cutlets. The camera was equipped with lens EZ F3.5–6.3 M. Zuiko ED, 12–50 mm with 24 mm, aperture f8 and with a circular polarizing filter. The 24 mm focal length and f8 aperture were chosen because, with this focal length and aperture, the vignetting phenomenon that could jeopardize the uniformity of illumination in the sensor is not detectable. A Macro Olympus OM T28 Double Flash was used with the heads placed opposite each other, very close to the lens axis and both pointed at 90° angle with the plane of the photography to obtain the most uniform intensity of light on the surface of the cutlets. Both flash heads had polarizing filters so that, in conjunction with the lens polarizing filter, the cross-polarization effect is obtained, to remove glare, specular highlights and better detail resolution as already stated in articles published in other areas where similar equipment in the same configuration as ours is used [21,22,23].

### 2.5. Muscle and Subcutaneous Fat Measurements

The cutlets were subjected to image analysis to determine muscle and subcutaneous fat measurements. The determination of tissue measurements was performed using Fiji software (ImageJ 1.49u) [24]. For tissue measurements, the first step is to convert the pixels to mm, using the rule placed beside the cutlet in the photo as scale. The measurements determined are for muscle—width—maximum width of LTL muscle, height—maximum height of LTL muscle, REA—rib eye area; and for subcutaneous fat, BFT—backfat thickness (Figure 1).

### 2.6. Marbling Fleck Features Extraction

To determine the marbling fleck features extraction, the image analysis was performed in the following steps—the first step was delimiting the region of interest (ROI), which represents the LTL (Figure 2a). For the selection of the ROI, care was taken not to include subcutaneous fat or intermuscular fat, which could be confused with the marbling flecks and lead to its overvaluation. After the ROI is converted into 8 bits in a grey scale image (Figure 2b), after five particles were identified unequivocally as marbling, we determined the gray level histogram to establish a threshold for marbling flecks. This threshold was set and the marbling flecks were highlighted (Figure 2c) and then segmented (Figure 2d). To remove small undesirable artefacts, the size particle must have at least a 10 pixels to be designated as a marbling fleck. The final step is the data extraction from the selected marbling fleck particles (Figure 2e). This data includes the number of marbling particles (NOParticles) and area of the marbling particles (Marb_area). Also, we determined the percentage of marbling as the relation between the area of the marbling particles with the area of the ROI (Marb_area%). The Fiji software (ImageJ 1.49u) [22] was used for image analysis of the marbling fleck. 

### 2.7. Statistical Analysis

This work intended to carry out a semi-quantitative (multivariate discrimination) and quantitative (multivariate regression) analysis by relating a dependent variable (IMF%, percentage of intramuscular fat of the Longissimus thoracis et lumborum muscle) with 13 independent variables (1 animal’s body parameter, 7 image analysis (IA) measurements and 5 CIELAB color measurements) that have been scaled and centered.

For discriminant semi-quantitative analysis, two supervised multivariate techniques were used and compared—linear discriminant analysis (LDA) and discriminant mixture analysis (MDA). LDA is a dimensionality reduction technique that allows us to obtain new functions, which are linear combinations of the independent variables, allowing maximum separation between the established groups and data projection into a lower dimensional-space [25,26]. MDA considers that each analyzed class is a Gaussian mixture of subclasses, where each data point has a probability of belonging to each class. Generally, MDA is used when there are variable classes and when there are more than two classes with responses that do not follow a pattern [25,26].

Cross-validation, such as K-fold cross-validation (internal validation), is used to assess the predictability of discriminant models, when the number of results in the data matrix is not high. The cross-validation with K-fold divides randomly the observations in the data set into k subgroups of approximately equal size. For comparison reasons, the K-fold groups were the same for the applied multivariate techniques. The first subgroup is treated as a validation set and the model is obtained using the rest. The process is suggested so that each subgroup is used in the validation. This methodology is robust because the validation is performed on a subset of samples that encompasses all groups that are involved in the classification [25,26]. In this work, cross-validation with 7 folds and 5 repetitions was used, allowing us to obtain 35 different models, and the global results were evaluated by the average accuracy (correct classifications).

The best predictive model was assessed on its ability to perform correct classifications (sensitivity, selectivity and accuracy). Sensitivity is the proportion of positive results out of the number of samples which were positive and specificity is the proportion of negatives among the truly negative ones (the smaller the number of false positives, the greater the specificity), is calculated in each group [25]. The average between sensitivity and specificity of each group corresponds to the balanced accuracy. Accuracy gives the overall model’s correct classifications, being the proportion of correct predictions to overall predictions. 

For regression analysis, two supervised multivariate techniques were used and compared—multiple linear regression (MLR) and polynomial support vector machine regression (SVMR-Poly). MLR allows us to obtain a linear equation (the model) between the explanatory variables (independent variables) and a response variable, the dependent variable [25,26]. SVMR-Poly is based on a polynomial kernel that represents the similarity of vectors (training samples) in a feature space over polynomials of the original variables, as transformed higher dimensional space, allowing learning of non-linear models [25,26,27]. The polynomial kernel feature space is equivalent to that of polynomial regression, but without the huge number of parameters to be learned.

Both models were evaluated in terms of the performance of prediction, using cross-validation K-folds (internal validation) by applying, as in the analysis of the previous section, 7 folds with 5 repetitions [26,28]. The 35 different models were evaluated by the average of the determination coefficients (R^2^) and root mean square errors (RMSE) values obtained from the adjustments between the values predicted by the models and the experimental ones.

The evaluation of the best model predictive capacity was made through the values of the slope and intercept (as well as, the respective confidence intervals) plus the coefficient of determination (R^2^), obtained from the linear relationship between the expected experimental and predicted values for the best-established model using data of K-folds cross validation. It is expected to have single linear regression parameters close to the theoretical values for the good predictive model—RSE (0), slope (1), the intercept (0) and the adjusted determination coefficient (1). Also, the confidence interval at 95% of the slope and intercept can be used to confirm that statistically they could be regarded as the theoretic values of “one” and “zero”, respectively [26].

All data processing and statistical methods were performed with the statistical program “open source” R version 4.02 GUI 1.72 (Vienna, Austria, Mac Catalina build) and RStudio version 1.3.959, using the following packages—caret [28] and e1071 [29] for SVM regression; gridExtra [30], ggplot2 [31] and scales [32] for data visualization; MASS [33] and mda [34] for discriminant analysis; and, psych [35] for basic descriptive statistics.

## 3. Results and Discussion

### 3.1. Data Matrix

Table 1 shows the minimum, maximum, median and coefficient of variation (CV) of the dependent variable (IMF%, intramuscular fat in percentage) and 13 independent variables—one body composition value (CW, carcass weight), five parameters from CIELAB color (L*—lightness, a*—yellowness, b*—redness, C*—Chroma and H*—Hue) and seven IA measurements (Width—maximum width of LTL muscle, Height—maximum height of LTL muscle, REA—rib eye area, BFT—back fat thickness, NOParticles—number of marbling particles, Marb_area—intramuscular fat in LTL area and Marb_area%—percentage of intramuscular fat in the LTL area) for each animal and sample. 

The carcass weight (CW) varying between 14.9 and 89.3 kg, was used as an independent variable representative of the animal maturity. The percentage of intramuscular fat (IMF%) is in the range of 0.62 to 2.08% of the body composition. This parameter corresponded to the model’s dependent variable and showed an acceptable variability between animals, which was desired to ensure variability in the independent data obtained through the analysis of muscle image and color analysis. 

These measurements were performed in order to obtain three concordant results, in general, with a percentage standard deviation (CV%) below 5%. Some measurements showed CV% values above 5% but, occasionally, they are samples that due to the variability of the color and the conformation of the muscle gave rise to more varied results. It has not been corrected because it is considered to be a real situation in the analysis and therefore represents intrinsic variability of the methodology. It was in the color variables that the highest number of measurements was obtained with CV% above 5%, reaching 15% (H* < 14%; b* < 15%) and in the case of the variable a*, 1 case with 20% and another with 30%.

Of the 20 Longissimus thoracis et lumborum muscle samples analyzed, a database was obtained with a dependent variable (IMF%) of 20 results and 13 independent variables with 39 measurements, having been performed in duplicate.

### 3.2. Semi-Quantitative Analysis

In order to verify the possibility of providing information on marbling to the consumers as a determinant factor to meat juiciness, a discriminant semi-quantitative data analysis was performed, which consists of establishing groups of intervals of intramuscular fat that are associated with levels of meat quality. Three semi-quantitative groups were established for the percentage of intramuscular fat (IMF%)—group 1, 0.6 to 1.1% (central value, 0.85%); group 2, 1.25 to 1.5% (central value, 1.38%); and, group 3, comprising values greater than 1.5% (central value, 1.79%). Considering these groups, 19 measurements are from group 1, eight measurements from group 2 and 12 measurements from group 3. Tyra and Zak [36] in a study with a Polish pig breed also divided the pig population into groups of breeds according to the levels of IMF. This effect of IMF as well the visual marbling assessment is reflected in the SEUROP grading system as verified by Ludwiczak et al. [37] in a novel approach for measuring pork marbling.

In this section, two classification models were studied, one linear and one non-linear, to semi-quantitatively predict the percentage of intramuscular fat in the LTL muscle using the CW, the IA measurements (Width, Height, REA, BFT, NOParticles, Marb_area and Marb_area%) and the CIELAB color measurements (L*, a*, b*, C* and H*). The models were—linear discriminant analysis (LDA) and mixture discriminant analysis (MDA). These two models are distinguished by the increasing order of complexity and were carried out with cross-validation K-folds (seven subsets of sample data with five repetitions; evaluation of 35 models using five different samples as test data in each model) to assess the predictive capacity inherent in the independent variables used. To evaluate the performance of LDA and MDA models, the average accuracy from the 35 tested models as well a, the best model’s accuracy and the balanced accuracy obtained for each of the three defined groups were used (Table 2). Figure 3 shows a graph of the distribution of the samples in the bi-dimensional space defined by the two linear discriminant functions and the boundary lines separating the zones defined for each group’s discrimination; and, a second graph, presenting the MDA results, in the same bi-dimensional space defined by the two discriminating functions, for comparison purpose.

Table 2 shows that LDA models have no satisfactory results in the cross-validation prediction from seven folds and five replications, verifying that, on average, the models have 68% correct classifications. This result can be explained by visualizing the LDA graph in Figure 3, where the division between groups 2 and 3 does not allow an evident separation between the respective samples. However, the best LDA model had an acceptable accuracy of 95% of correct classifications, related to the balanced accuracy results—100% of correct classifications in the group 1 (100% for sensitivity and specificity), 97% of correct classifications in the group 2 (related to 100% of sensitivity and 94% of specificity) and 92% of correct classifications in the group 3 (related to 83% of sensitivity and 100% of specificity). These results show that 2 samples from group 3 were incorrectly classified as group 2.

Better results were obtained with the MDA models that allowed 100% of correct classifications in data used for obtaining the models, showing that all groups presented sensitivity and specificity of 100% (Figure 3). The decreasing order of the variables’ importance to the discriminant model was—BFT; CW; H*; REA; Height; b*; Marb_area; Width; NOParticles; a*; Marb_area%; L*; and, C*.

Overall, the results obtained show that the relationship between the dependent variable IMF% and the 13 independent variables is not linear, being MDA, a suitable multivariate technique to establish a semi-quantitative classification model of IMF% based on carcass weight, CIELAB color and IA measurements. Other authors in a study with two pig breeds in different production systems also showed that it was possible to generate high variability in the technological qualities and sensory attributes such as meat CIELAB color, tenderness and juiciness to establish correlations between those meat quality traits and marbling or intramuscular fat [38].

### 3.3. Quantitative Analysis

A more complex task is regression analysis to predict the percentage of intramuscular fat in the LTL muscle using the animal’s carcass weight, the 7 IA measurements and the five CIELAB color measurements. For this purpose, MLR, a linear regression technique, and SVMR-Poly, a nonlinear regression technique, were applied. Table 3 shows the results obtained from cross-validation and the best models selected for each of the multivariate regression techniques.

In general, the MLR models showed overall cross-validation results of low determination coefficients and high RMSE values, demonstrating difficulty in obtaining good adjustments between IMF% and the 13 independent variables. The best model gave a linear relationship between the predicted values and the experimental values of IMF% that only explains 86% of the variability of the data used and has a high RSE value. It was also found that the intercept at the origin is not close to zero (nor is it defined in the respective confidence interval) and the slope is low and different from one (not within the confidence interval). 

As can be seen in Figure 4, the graph of the relationship between the predicted values by the MLR model and the experimental values of IMF% showed high variability in relation to the linear fitting. These results gave evidence that data do not follow a linear relationship.

SVMR-Poly was used as a nonlinear supervised multivariate regression technique to verify if it could modulate the data. The overall SVMR-Poly cross-validation results (Table 3) showed acceptable higher determination coefficients and lower RMSE values. Also, the best model (model’s parameters optimized to degree of three with a scale factor of 0.1 and a cost value of one) could represent 99.9% of the data variability (resulting in a low RSE value) and present good predictive abilities, as can be verified by the slope value near to one (within the confidence interval) and an intercept of zero, for the linear relationship between the predicted values by SVMR-Poly model and the experimental values of IMF%. The intercept was not considered in the linear fitting (data not shown in Table 3) because it was verified as not having significance in the linear model (*p*-value > 0.05). The obtained linear relationship is represented in Figure 4, where it can be seen that SVMR-Poly model, a nonlinear method, allowed us to obtain a proper fitting between IMF% and the 13 independent variables. The best SVMR-Poly model, with the decreasing order of the importance of variables to the discriminant model was—BFT; CW; REA; b*; Width; Marb_area; Height; L*; NOParticles; a*; C*; and, Marb_area%. Only the H* parameter was not considered important for the model. 

The results found in the present work (Rc^2^ = 0.999, Rcv^2^ = 0.88 for the best SVMR-Poly model, being the determination coefficients of calibration and cross-validation, respectively) show accuracy in the estimation of pig marbling and intramuscular fat, similar to that reported by other authors with imaging and spectroscopic techniques. For example, Huang et al. [13], who used a pattern recognition technique in Red-Green-Blue images of fifty-three fresh chops, reported high correlation coefficients of calibration and validation (Rc^2^ = 0.88, Rv^2^ = 0.88) for pork marbling assessment. More recently, Liu et al. [16], using a computer vision system to predict IMF% in 85 pork loins from regression models which included eighteen color components, including L * a * b *, presented an accuracy of 0.63 for stepwise and 0.75 for support vector machine. For its part, Font-i-Furnols et al. [11], using computed tomography images of 365 pork loins to estimate intramuscular fat, verified the best prediction of IMF was obtained by linear regression when data from two tomograms were used (R^2^ = 0.83 and RMSE = 0.46% of cross-validation). These results with imaging techniques are comparable to others obtained with spectroscopic techniques. For example, Huang et al. [14], who examined 144 pork loin samples with hyperspectral imaging, found that the best MLR models to estimate the IMF content show a high predictive ability (Rc^2^ = 0.92, Rcv^2^ = 0.90 and Rfv^2^ = 0.69; calibration, cross-validation and full validation determination coefficients). These results are comparable to other similar studies predicting the intramuscular fat content of pigs using hyperspectral images [39]. However, in a recent study, Andersen et al. [40] in which they studied 122 LTL samples to estimate the IMF achieved modest results with NIR and fluorescence spectroscopy techniques, with Rcv^2^ of 0.57 and 0.18, respectively. However, with the Raman technique, the results were more promising to predict intramuscular fat and using partial least squares regression, as the Rcv^2^ was 0.73. The results found by those authors and also in the present work show that the imaging techniques to evaluate meat quality features are a solid path. In addition, rapid technological development will improve and challenge current methods of image acquisition and pattern recognition for the evaluation of meat quality [41]. It is expected that in the short term, faster, smaller and inexpensive hardware will be used to improve the image acquisition for the meat industry and research [41,42]. In addition, it is expected that smartphone-derived picture image analysis can be used as a simple and quick way to assess the quality and safety of meat [43].

## 4. Conclusions

In this work, a set of Longissimus thoracis et lumborum (LTL) muscle of Bísaro pigs with a wide weight range of the carcasses allowed us to achieve robust predictive models. It was possible to predict meat juiciness (defined by groups of intervals of intramuscular fat) and intramuscular fat percentage using 13 independent variables comprised carcass weight, five parameters from CIELAB color and seven IA measurements. The results showed that data fitting demanded nonlinear models be used with mixture discriminant analysis in the semi-quantitative analysis and support vector machine regression with a polynomial kernel for the quantitative analysis.

The importance of this work results from the demonstration that the application of technologies can contribute to better efficiency in the evaluation of meat quality. Mainly, it shows the possibility of obtaining the percentage of intramuscular fat, which is a time-consuming and expensive chemical analysis, by processing the photographic image and the color of that muscle. No similar study was found in the bibliography for comparative purposes, emphasizing that the effectiveness of the methodology should be valid in future works to obtain a more extensive data matrix. Finally, this work shows potential application to Bísaro pigs in small industrial unites and that in future work we will develop new color image metrology in a larger sample to deepen the knowledge developed here.

## Figures and Tables

**Figure 1 foods-10-00143-f001:**
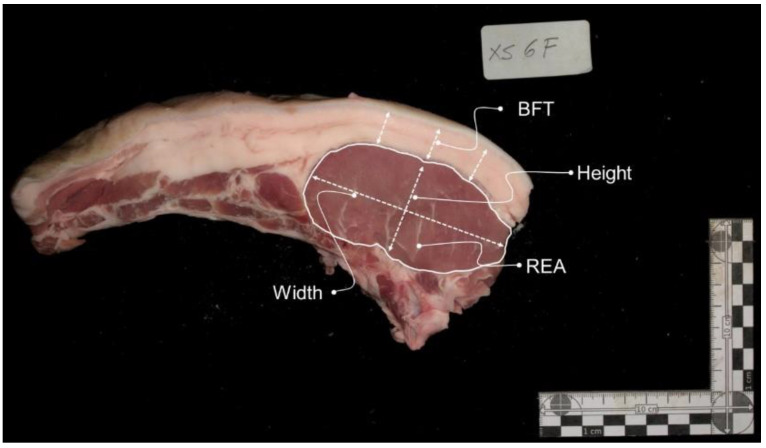
Image illustrating the muscle and subcutaneous fat measurements. LTL—Longissimus thoracis et lumborum; width—maximum width of LTL muscle; height—maximum height of LTL muscle; REA—rib eye area; BFT—backfat thickness.

**Figure 2 foods-10-00143-f002:**
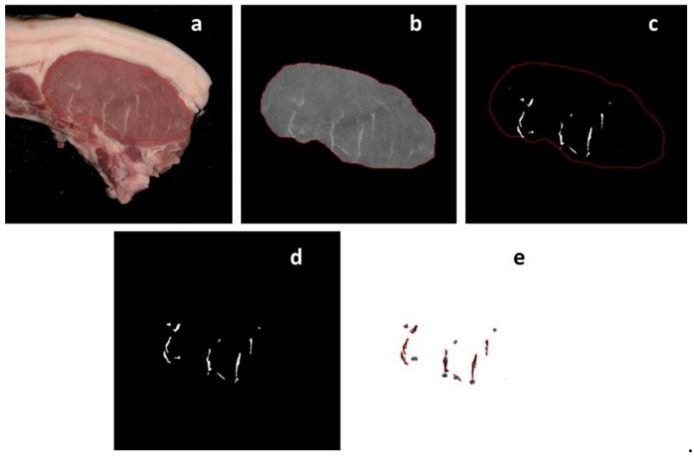
Steps of image analysis for the marbling fleck determination. (**a**) Original cutlet image with the region of interest (ROI); (**b**) clear outside the ROI and transform in a gray scale 8-bit image; (**c**) marbling fleck isolation after application of a threshold in a gray scale 8-bit image; (**d**) marbling fleck particles segmentation; and (**e**) numerical data extract.

**Figure 3 foods-10-00143-f003:**
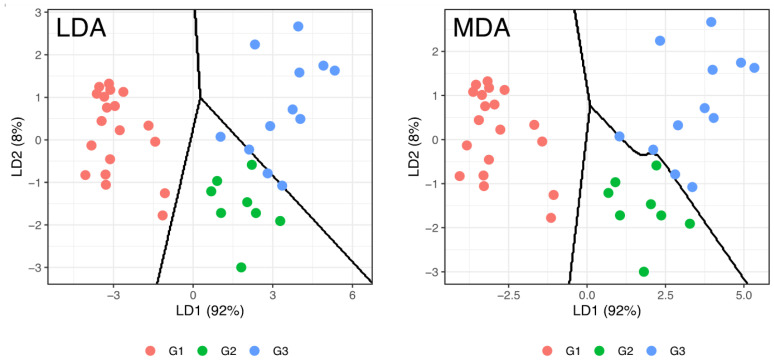
Representation of LDA and MDA discrimination of the samples grouped semi-quantitatively by the percentage of intramuscular fat and the boundary lines separating the zones defined for each group’s discrimination in the bi-dimensional space defined by the two linear discriminant functions. LDA—linear discriminant analysis; MDA—mixture discriminant analysis; LD1—first linear discriminant function; LD2—second linear discriminant function; IMF%—percentage of intramuscular fat; G1—IMF% group 1, 0.6 to 1.1% (central value, 0.85%); G2—IMF% group 2, 1.25 to 1.5% (central value, 1.38%); G3—IMF% group 3, comprising values greater than 1.5% (central value, 1.79%).

**Figure 4 foods-10-00143-f004:**
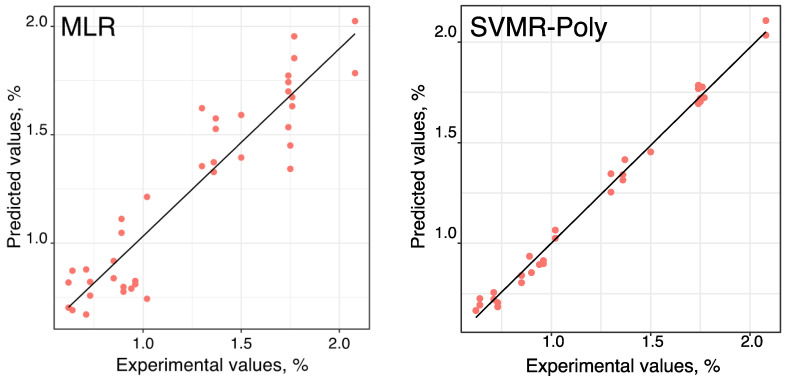
Representation of the linear relationship between the predicted values by MLR and SVMR-Poly models and the experimental values of IMF%. MLR—multiple linear regression; SVMR-Poly—polynomial support vector machine regression; IMF%—percentage of intramuscular fat.

**Table 1 foods-10-00143-t001:** Global results for all variables considered in this work.

Parameters	Minimum	Maximum	Median	CV (%)
Dependent variable				
IMF%	0.62	2.08	1.30	36.0
Independent variables				
Animal body variable				
CW (kg)	14.9	89.3	25.5	67.3
CIELAB color variables				
L*	44.8	61.9	55.1	7.0
a*	0.99	13.6	3.40	71.1
b*	6.84	14.6	10.9	18.5
C*	9.00	17.0	12.7	17.1
H*	34.1	83.8	73.4	23.2
IA variables				
Width (cm)	5.3	12.0	8.16	18.6
Height (cm)	2.74	8.75	4.45	28.0
REA (cm^2^)	11.7	55.0	23.7	40.4
BFT (cm)	0.39	6.05	1.30	65.1
NOParticles	16	59	27	34.5
Marb_area (mm^2^)	0.14	2.03	0.38	71.1
Marb_area%	1.03	3.92	1.75	36.1

IMF—intramuscular fat; CW—carcass weight; L*—lightness; a*—yellowness; c*—redness; C*—Chroma; H*—Hue; Width—maximum width of LTL muscle; Height—maximum height of LTL muscle; REA—rib eye area; BFT—back fat thickness; NOParticles—number of marbling particles; Marb_area– intramuscular fat in LTL area; Marb_area%—intramuscular fat % in LTL area.

**Table 2 foods-10-00143-t002:** Evaluation parameters of the LDA and MQA models.

Parameter	LDA	MDA
Average cross-validation results		
Accuracy	0.68 ± 0.17	0.92 ± 0.12
Best model prediction capability		
Accuracy	0.95 ± 0.03	1.00
CI_Accuracy_	[0.83, 0.99]	[0.91, 1.00]
*p*-value	<0.001	<0.001
G1 balanced accuracy	1.00	1.00
G2 balanced accuracy	0.97	1.00
G3 balanced accuracy	0.92	1.00

LDA—linear discriminant analysis; MDA—mixture discriminant analysis; CI—95% confidence interval; IMF%—percentage of intramuscular fat; G1—IMF% group 1, 0.6 to 1.1% (central value, 0.85%); G2—IMF% group 2, 1.25 to 1.5% (central value, 1.38%); G3—IMF% group 3, comprising values greater than 1.5% (central value, 1.79%).

**Table 3 foods-10-00143-t003:** Evaluation parameters of the MLR and SVM models.

Parameter	MLR	SVMR-Poly
Average cross-validation results		
Rcv^2^	0.76 ± 0.13	0.88 ± 0.12
RMSE	0.26 ± 0.08	0.18 ± 0.11
Best model prediction capability		
Rc^2^_Adjusted_	0.86	0.999
RSE	0.16	0.04
*p*-value	<0.001	<0.001
Slope	0.86 ± 0.06	0.993 ± 0.005
CI_Slope_	[0.75, 0.98]	[0.982, 1.000]
Intercept	0.17 ± 0.07	ns
CI_Intercept_	[0.02, 0.32]	na

ns—not significant; na—not available; CI—95% confidence interval; Rc^2^—R-squared, determination coefficient of calibration; Rcv^2^—R-squared, determination coefficient of cross-validation; RMSE—root mean squared error; RSE—residual standard error; MLR—multiple linear regression; SVM—support vector machine; SVMR-Poly—polynomial support vector machine regression.

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
