# Peer review of "Intramuscular Fat Prediction Using Color and Image Analysis of Bísaro Pork Breed"

_foods, 2021, doi:10.3390/foods10010143_

Round 1

Reviewer 1 Report

Well-presented article that addresses the classification of meat by its intramuscular fat content and its relationship with juiciness. These are aspects of great interest and relevance of the quality of pork. It uses a very wide range of weights, but the work is of interest and contributes to generating knowledge in little-studied traditional breeds, using relatively simple and affordable predictive techniques. Although the number of animals used is a bit low, we recommend the publication of the article with a minor revision.

Specific comments

L77: the number of animals is low.
L79: the prediction is made over a very wide weight range, which facilitates a good prediction. In practice, the method would be useful within a much more restricted weight range of around 100kg.
L90: define measuring points exactly.
L97, 98: the methodology for extraction of intramuscular fat is not clear. If done using a soxtec, what is the Folch method used for?
L99: the authors should explain how they cleanse the muscle of intramuscular fat...
L130: should explain how and what threshold is applied to define marbling flecks.
L148: what does the use of all CIELAB predictor variables contribute, when two of them are a combination of the other two?
L159: have the authors considered using a training and a validation sets instead of cross validation?
L236: juiciness is not only dependent on intramuscular fat content.
L357: in the conclusions it should be reflected that “it is possible to predict meat juiciness”, when the weight range of the carcass is very wide. Not for nothing BFT and CW are the variables of greatest importance for the best SVM-Poly model.

Author Response

Responses to reviewers

The authors would like to express gratitude and appreciation to the attention reviewers dedicated to this manuscript. We considered comments and suggestions carefully to address them, as described in detail in the following replies.

Review 1

Well-presented article that addresses the classification of meat by its intramuscular fat content and its relationship with juiciness. These are aspects of great interest and relevance of the quality of pork. It uses a very wide range of weights, but the work is of interest and contributes to generating knowledge in little-studied traditional breeds, using relatively simple and affordable predictive techniques. Although the number of animals used is a bit low, we recommend the publication of the article with a minor revision.

Specific comments

L77: the number of animals is low. 

Reply: The Bisaro is a local breed with farms with a reduced number of animals, and it is very challenging to obtain samples with a high number as usually seen in works with animals of commercial lines.

L79: the prediction is made over a very wide weight range, which facilitates a good prediction. In practice, the method would be useful within a much more restricted weight range of around 100kg. 

Reply: This work provides important information for other ranges of carcass weight below 100 kg that are traditionally consumed as "suckling pig" for example or even above 100 kg to produce some traditionally PDO processed produts.

L90: define measuring points exactly.

Reply: The points on muscle for color measurements are normally taken aleatory. The text was modified.

L97, 98: the methodology for extraction of intramuscular fat is not clear. If done using a soxtec, what is the Folch method used for? 

Reply: The Folch method was used for GC injection to assess the fatty acid profile. As these data were not used in this manuscript, we removed this reference.

L99: the authors should explain how they cleanse the muscle of intramuscular fat... 

Reply: It is an error. The fat cleaned was the intermuscular fat. The text was corrected.

L130: should explain how and what threshold is applied to define marbling flecks. 

Reply: The authors are grateful for the comment, and a new text (lines 29-31) was introduced.

L148: what does the use of all CIELAB predictor variables contribute, when two of them are a combination of the other two? 

Reply: It is not the same in terms of CIELAB approach. a * and b * are color coordinates, since C * an h * are the color attributes. The former allow the assessment of redness and yellowness, respectively, since the latter are the true assessment of color in terms of chroma and tone

L159: have the authors considered using a training and a validation sets instead of cross validation? 

Reply: Yes, but due to low number of samples it was decided that cross-validation was the best procedure. Future work will contemplate an external validation (test data group).

L236: juiciness is not only dependent on intramuscular fat content. 

Reply: We agree with the referee. We changed juiciness for marbling. In reality the information provided to consumers is the fat infiltrated on muscle.

L357: in the conclusions it should be reflected that “it is possible to predict meat juiciness”, when the weight range of the carcass is very wide. Not for nothing BFT and CW are the variables of greatest importance for the best SVM-Poly model.

Reply: Regardind this issue, new sentence was introduced.

Reviewer 2 Report

The paper deals with using color analysis and a classical pattern recognition method to discriminate between intramuscular fat /meat juiciness intervals in a pig species.

While the application is very relevant and well-studied already, the treatment in this paper is simplistic, especially regarding image metrology, impacts of small sample sizes, image analysis, and conclusions. An unnecessary large number of outputs from computing packages used is provided, while the assumptions and input are not discussed.

On the overall approach, it is reasonable, but ignores the benefits from extracting meat texture (e.g. [1]), using color scanning cameras, alternative classification methods such as neural networks (e.g. [2]), and proper color image metrology [3].

The training sample size in terms of exact number of labelled color images is not specified; is only specified the number of carcasses (20). Is missing as well the delay between slaughter and cooling, which affects some of the parameters of interest. The 5 parameters acquired from CIEL colors do not have the exact and needed number of sample points on the muscles.

There is missing details on the chemical analysis sample sizes. It is reminded that, if chemical analysis is used for truth determination, there are measurement errors on the chemical analysis outputs as well, which are not discussed.

There is yet another calibration issue. The spectrometric color determination and the image analysis use two different sensors, which is strange as there are on the market scanning spectrophotometers of various kinds. The result is that cross calibration of those two sensors must be accounted for.

Yet another image calibration issue is the absence of proper uniform illumination; actually, the use of two flash directions and polarization introduces illumination pattern oddities, and it is not stated if a zero-background image was subtracted.

If camera image optics are left fixed, Section 2.6 does not state how blurring due to muscle thickness vs. 30 cm line of sight is handled. Likewise, the physical pixel size is not specified at muscle surface.

Just to state in Section 2.5 “For both, the IA was performed using Fiji 113 software (ImageJ 1.49u, https://imagej.nih.gov/ij/) [22]” is not acceptable in an academic paper. The image acquisition method with its exact parameters (including sample size corrections etc.) must be specified. A similar remark holds for image analysis in Section 2.7. For normal readers the abbreviation IA must be defined.

In Section 2.7, the reference to 8-bit image is imprecise; are there 8 bits for each color component, or only 8 bits in a grey scale image?

The sentence “This technique assumes the equality of covariance matrix between classes” is wrong, unless the authors have used this much simplified assumption.

The description of cross-validation is too much simplified, as long as sample sizes (see previous remarks) are not provided, and statistical distributions in each class are not assumed and estimated. As a consequence, results on confidence intervals are conjectures. It is assumed that the paper assumes normal distributions, but this is not stated nor verified from the data.

The sentence “All data processing and statistical methods were performed with the statistical program "open 198 source" R version 4.02 GUI 1.72 (Mac Catalina build) and RStudio version 1.3.959, using the following 199 packages: caret [26], e1071 [27], gridExtra [28], ggplot2 [29], MASS [30], mda [31], psych [32] and 200 scales [33].” Is not acceptable in an academic paper, where all parameters used must be specified as they impact the conclusions in Section 3.

The statement “with the objective of verifying which 245 multivariate methodology is most suitable to semi-quantitatively predict the percentage of 246 intramuscular fat” is oversimplistic, as it has forgotten the basic fact that all is driven by the input data (including volume of such) and their calibration, and also the supervised and unsupervised sample sizes in this study are still rather small.

Thus, the conclusions must be revised to accommodate for all the factors above, as this experiment does not necessarily scale up or apply to other data and settings.

Regarding the English language, there is a tendency to write too long sentences; so, cut the sentences or use punctuation. Also, the terms used are often not precise enough, or overblown (“polynomial support vector machine “is just a polynomial discriminant function). Just in the abstract:

-In abstract: “Both ….”: which both?

-In abstract: “the obtained model…”. Ambiguous as there is apparently no quantitative model, but a set of classification rules!

-In abstract: “optimized to degree of 3”?? You mean that the estimated discriminant function is a polynomial of degree 3?

-Discriminant mixture analysis:?? You mean “discriminant analysis” which is often used in pattern recognition and multivariate statistics?

The references are extensive for the specific application of interest, but weak on image acquisition and pattern recognition.

[1] https://www.researchgate.net/publication/329409492_Classification_of_pork_and_beef_meat_images_using_extraction_of_color_and_texture_feature_by_Grey_Level_Co-Occurrence_Matrix_method

[2] https://iopscience.iop.org/article/10.1088/1757-899X/434/1/012072/meta

[3] https://www.nist.gov/laboratories/tools-instruments/color-and-appearance-metrology-facility

Author Response

Responses to reviewers

The authors would like to express gratitude and appreciation to the attention reviewers dedicated to this manuscript. We considered comments and suggestions carefully to address them, as described in detail in the following replies.

Reviewer 2

The paper deals with using color analysis and a classical pattern recognition method to discriminate between intramuscular fat /meat juiciness intervals in a pig species.

While the application is very relevant and well-studied already, the treatment in this paper is simplistic, especially regarding image metrology, impacts of small sample sizes, image analysis, and conclusions. An unnecessary large number of outputs from computing packages used is provided, while the assumptions and input are not discussed.

On the overall approach, it is reasonable, but ignores the benefits from extracting meat texture (e.g. [1]), using color scanning cameras, alternative classification methods such as neural networks (e.g. [2]), and proper color image metrology [3].

Reply: The authors are grateful for the indication of these technologies that will be considered in future work and as equipment to acquire for our laboratory. This is an area in which we are a keen interest and comments, and indications for another type of image metrology are welcome.

The training sample size in terms of exact number of labelled color images is not specified; is only specified the number of carcasses (20). Is missing as well the delay between slaughter and cooling, which affects some of the parameters of interest.

Reply: A new text was introduced (L81-82 then the carcass was placed in a cooling chamber).

The 5 parameters acquired from CIEL colors do not have the exact and needed number of sample points on the muscles.

Reply: The collection of color variables CIELab was performed on the muscle at random so that a good representation of the muscle color itself was achieved.

There is missing details on the chemical analysis sample sizes. It is reminded that, if chemical analysis is used for truth determination, there are measurement errors on the chemical analysis outputs as well, which are not discussed.

Reply: for the determination of intramuscular fat, subcutaneous and intermuscular fats were removed and this sample was analyzed according to a very well established procedure that has been used as reference for this type of work.

There is yet another calibration issue. The spectrometric color determination and the image analysis use two different sensors, which is strange as there are on the market scanning spectrophotometers of various kinds. The result is that cross calibration of those two sensors must be accounted for.

Reply: The sensors used had different objectives. One was to determine the CIELab color in which we use the Minolta CM-2006d spectrophotometer equipment, which is the equipment widely used in food science to measure the color of meat and certified by the CIEL (Commission Internationale de l'Eclairage). The other sensor is related to the acquisition of images from which information about Marbling data was obtained. As mentioned above, the authors are grateful for the suggestion of new equipment, but in this case we try to carry out the work with the equipment that already exists in our laboratory.

Yet another image calibration issue is the absence of proper uniform illumination; actually, the use of two flash directions and polarization introduces illumination pattern oddities, and it is not stated if a zero-background image was subtracted.

Reply: We appreciate the comment and a sentence was introduced and now reads as (lines xx-xx): we introduced new text on this subject. The camera was equipped with lens EZ F3.5-6.3 M. Zuiko ED, 12-50 mm with 24 mm, aperture f8 and with a circular polarizing filter. The 24mm focal length and f8 aperture were chosen because, with this focal length and aperture, the vignetting phenomenon that could jeopardize the uniformity of illumination in the sensor is not detectable. A Macro Olympus OM T28 Double Flash was used with the heads placed opposite each other, very close to the lens axis and both pointed at 90º angle with the plane of the photography to obtain the most uniform intensity of light on the surface of the cutlets. Both flash heads had polarizing filters so that, in conjunction with the lens polarizing filter, the cross-polarization effect is obtained, to remove glare, specular highlights and better detail resolution as already stated in articles published in other areas in medicine where similar equipment in the same configuration has ours is used [1-3].

Subtraction of the background image was not considered since after the first step, an ROI was obtained with which we worked until the extraction date.

1 - Hanlon, K. L. Cross-polarised and parallel-polarised light: Viewing and photography for examination and documentation of biological materials in medicine and forensics. J. Vis. Commun. Med. 2018, 41, 3–8. DOI: 10.1080/17453054.2018.1420418

2 - He, W.-H., Park, C. J., Byun, S., Tan, D., Lin, C. Y., Chee, W. Evaluating the relationship between tooth color and enamel thickness, using twin flash photography, cross-polarization photography, and spectrophotometer. J. Esthet. Restor. Dent. 2020, 32, 91–101. DOI: 10.1111/jerd.12553

3 - Villavicencio-Espinoza, C. A., Narimatsu, M. H., Furuse, A. Y. Using cross-polarized photography as a guide for selecting resin composite shade. Oper. Dent. 2018, 43, 113–120. DOI: 10.2341/16-227-T

If camera image optics are left fixed, Section 2.6 does not state how blurring due to muscle thickness vs. 30 cm line of sight is handled. Likewise, the physical pixel size is not specified at muscle surface.

Reply: We appreciate the comment and a sentence was introduced and now reads as (lines 90-100): we introduced new text on this subject. The pieces were all cut to the same thickness, and as such, the effect of the distance from the camera to the sample is always similar.

Just to state in Section 2.5 “For both, the IA was performed using Fiji 113 software (ImageJ 1.49u, https://imagej.nih.gov/ij/) [22]” is not acceptable in an academic paper. The image acquisition method with its exact parameters (including sample size corrections etc.) must be specified. A similar remark holds for image analysis in Section 2.7. For normal readers the abbreviation IA must be defined.

Reply: The authors welcome the comment and an amendment was made. Please see Lines 112-114, 125 and 137-138.

In Section 2.7, the reference to 8-bit image is imprecise; are there 8 bits for each color component, or only 8 bits in a grey scale image?

Reply: This sentence was revised and now reads as (line 144): The text is changed it is 8 bits in a grey scale image

The sentence “This technique assumes the equality of covariance matrix between classes” is wrong, unless the authors have used this much simplified assumption.

Reply: It was an error. The phrase was removed.

The description of cross-validation is too much simplified, as long as sample sizes (see previous remarks) are not provided, and statistical distributions in each class are not assumed and estimated. As a consequence, results on confidence intervals are conjectures. It is assumed that the paper assumes normal distributions, but this is not stated nor verified from the data.

Reply: MDA is a variant of discriminant analysis, in which classes are not modeled as Gaussian distributions, as is common practice for LDA, but rather classes are modeled as a mixture of Gaussian distributions.

 LDA and MDA make assumptions about data. For instance, in MDA, there are classes, and each class is assumed to be a Gaussian mixture of subclasses, where each data point has a probability of belonging to each class. LDA is based on the assumption that each class can be modeled by a Gaussian distribution and that all the classes share the same covariance matrix.

However, even if the assumptions are not fully complied, it does not mean that it cannot be applied as the predictive results are those that allow us to evaluate the robustness of the model. So, in this work, we gave particular importance to the predicted data in the evaluation of the models instead of testing the assumptions to the data.

If the reviewer considers this point important for the present work, then we will introduce new data into the manuscript.

The sentence “All data processing and statistical methods were performed with the statistical program "open 198 source" R version 4.02 GUI 1.72 (Mac Catalina build) and RStudio version 1.3.959, using the following 199 packages: caret [26], e1071 [27], gridExtra [28], ggplot2 [29], MASS [30], mda [31], psych [32] and 200 scales [33].” Is not acceptable in an academic paper, where all parameters used must be specified as they impact the conclusions in Section 3.

Reply: The references presented are in accordance with those requested by the authors of the packages. Some packages were used to data processing, visualization and statistical analysis. However, to clarify how they were used, a new text was introduced in which a brief description of how they were used at work was made. The text was changed to:

“All data processing and statistical methods were performed with the statistical program "open source" R version 4.02 GUI 1.72 (Mac Catalina build) and RStudio version 1.3.959, using the following packages: caret [28] and e1071 [29] for SVM regression; gridExtra [30], ggplot2 [31] and scales [32] for data visualization; MASS [33] and mda [34] for discriminant analysis; and, psych [35] for basic descriptive statistics.”

The statement “with the objective of verifying which multivariate methodology is most suitable to semi-quantitatively predict the percentage of intramuscular fat” is oversimplistic, as it has forgotten the basic fact that all is driven by the input data (including volume of such) and their calibration, and also the supervised and unsupervised sample sizes in this study are still rather small.

Reply: Although it was not mentioned, in data treatment, several discriminant techniques were tested, presenting in this work the MDA method that best fits the data and LDA for comparison purposes. The phrase was changed.

Thus, the conclusions must be revised to accommodate for all the factors above, as this experiment does not necessarily scale up or apply to other data and settings.

Reply: A new text was introduced in conclusion. Please see lines 395 to 396. Finally, this work shows potential application to Bísaro pigs in small industrial unites and that in future work we will develop new color image metrology in a larger sample to deepen the knowledge developed here.

Regarding the English language, there is a tendency to write too long sentences; so, cut the sentences or use punctuation.

Reply: All manuscript has been revised to improve its clarity.

Also, the terms used are often not precise enough, or overblown (“polynomial support vector machine “is just a polynomial discriminant function). Just in the abstract:

Reply: This technique can be used for both qualitative and quantitative analysis. In this work, it was used to obtain a regression model and, therefore, it should not be considered as a discrimination technique. To better clarify this point, the term regression was included when talking about the SVM technique.

-In abstract: “Both ….”: which both?

Reply: The word “Both” referred to the semi-quantitative and quantitative models, referred to in the first paragraph of the abstract. The text was changed.

-In abstract: “the obtained model…”. Ambiguous as there is apparently no quantitative model, but a set of classification rules!

Reply: As referred in the first paragraph of the abstract two models were obtained in this work using as independent variables the Bísaro pigs carcass weight and parameters from color and image analysis of its Longissimus thoracis et lumborum (LTL) muscle:

- a discriminant model to predict meat juiciness (semi-quantitative analysis using groups of intervals of intramuscular fat) of the Longissimus thoracis et lumborum (LTL) muscle;

- a regression model for intramuscular fat prediction (quantitative analysis) in Longissimus thoracis et lumborum (LTL) muscle.”

So, to clarify this point the text was revised in order to distinguish the well between the “discriminant” and the “regression” models. As well, the abbreviation SVM-Poly was changed to SVMR-Poly.

-In abstract: “optimized to degree of 3”?? You mean that the estimated discriminant function is a polynomial of degree 3?

Reply: The SVM algorithm is versatile since it supports linear/nonlinear classification, and linear/nonlinear regression. For regression it reverses the objective of discrimination; instead of having a largest boundary separating classes, SVM Regression tries to fit as many experimental points as possible on a narrow boundary. The degree parameter controls the flexibility of the decision boundary. Higher degree kernels yield a more flexible decision boundary. The polynomial kernel function was established with degree 3, but it should no be considered as a third-degree polynomial function. The text in the abstract was changed to clarify this issue.

-Discriminant mixture analysis:?? You mean “discriminant analysis” which is often used in pattern recognition and multivariate statistics?

Reply: Discriminant analysis includes the multivariate statistical techniques used to develop discriminant functions that relate independent variables with the categories of the dependent variable. For this, there are several linear and non-linear supervised techniques as like the two applied in this work: linear discriminant analysis (LDA) and discriminant mixture analysis (MDA). So, discriminant mixture analysis is a discriminant analysis by gaussian mixtures (as already referred in page 5, lines 1-5).

The references are extensive for the specific application of interest, but weak on image acquisition and pattern recognition.

Reply: A text was introduced. Please see Lines 362-369.

[1]https://www.researchgate.net/publication/329409492_Classification_of_pork_and_beef_meat_images_using_extraction_of_color_and_texture_feature_by_Grey_Level_Co-Occurrence_Matrix_method

[2] https://iopscience.iop.org/article/10.1088/1757-899X/434/1/012072/meta

[3]https://www.nist.gov/laboratories/tools-instruments/color-and-appearance-metrology-facility
